



# The operational 3DEnVar data assimilation scheme for the Météo-France convective scale model AROME-France

Pierre Brousseau[1,*], Valérie Vogt[1,*], Etienne Arbogast[1], Maud Martet[1], Guillaume Thomas[1], and Loïk Berre[1]

[1]CNRM, Université de Toulouse, Météo-France, CNRS, Toulouse, France
[*]These authors contributed equally to this work.

**Correspondence:** Pierre Brousseau (pierre.brousseau@meteo.fr)

**Abstract.** Since October 2024 the Météo-France operational convective scale model AROME-France uses a 3DEnVar data assimilation (DA) scheme in order to improve the performances of severe weather prediction. This paper describes the configuration and the evaluation of this 3DEnVar scheme. It summarises the work carried out to configure this scheme in an operational context, with the inherent constraints of numerical robustness, compatible execution times and, of course, correct

performances of the forecasts produced. The adjustment of horizontal and vertical localization, inflation, hybridization and the use of Incremental Analysis Update (IAU) are studied in sensitivity experiments, and the impact on the spin-up is investigated. A configuration, and its variation with IAU, are thus defined and evaluated using different scores over a long period and on different severe meteorological situations (winter storm, fog and High Precipitating Events): they largely outperform the operational 3D-Var. The version with IAU has been implemented in the new operational version of the AROME-France assimilation

system.

## 1 Introduction

For fifteen years now, AROME-France has been the Météo-France operational Numerical Weather Prediction (NWP) System for convective scales. Since the beginning, this model, as many of its equivalents (Gustafsson et al., 2018), has been equipped with a Data Assimilation (DA) system to benefit from initial conditions that contain information at scales that the model

resolves. Using the same principles as for lower resolution NWP systems, this DA system combines regional observations with a previous high resolution model forecast, to produce a 3D analysis at the model resolution. The first version (Seity et al., 2011) used a 2.5 km horizontal resolution in a 3-hour continuous assimilation cycle, based on a 3D-Var scheme. Numerous improvements to the AROME-France system have been implemented regularly, approximately once a year, and two major ones occurred in April 2015 (Brousseau et al., 2016) : the increase of vertical and horizontal model resolutions and the increase of

the DA cycle frequency (from 3 to 1-hour period cycle).

This DA system has demonstrated its ability to provide very useful information to improve general scores of forecast performances. In many situations, it also helps forecasters to take right decisions during high impact events (High Precipitating Events, fog, storm... ). However, it suffers from two major flaws :





– the use of climatological background error statistics (the so-called **B** matrix) modelled using Berre (2000) formulation

in spectral space under the assumption of horizontal homogeneity and isotropy (Brousseau et al., 2011).

– the lack of temporal dimension within the assimilation window in the 3D scheme, which restricts the frequency of assimilated observations to the assimilation cycle frequency (the test of the First Guess at Appropriate Time (FGAT) version does not show any improvements).

Regarding the first limitation, numerous studies have tried to introduce some time-dependency in the AROME-France frame-
30 work, using either update of the different coefficients implied in the covariance formulation (Brousseau et al., 2012; El-Said et al., 2022) or geographical masks to deal with dependencies to different meteorological phenomena (Montmerle and Berre, 2010; Ménétrier and Montmerle, 2011). However, the impact on the performances remains very weak, compared to the increase of the numerical cost of such implementations. To go further on these aspects, larger improvements can be expected from ensemble-based formulations, such as 3DEnVar implemented at NCEP (Wu W. and Lin, 2017) and Local Ensemble
Transform Kalman Filter (LETKF) used in operations at DWD (Schraff et al., 2016). Concerning the second limitation, several NWP operational centers (UK Met Office, Ballard et al. (2015), Japan Meteorological Agency, Honda and Sawada (2008)) currently employ a 4D-Var scheme, using an analysis increment at a lower horizontal resolution than the non-linear model, in order to allow numerical cost acceptable for an operational use (see also Gustafsson et al. (2012) for a previous regional 4D-Var application with analysis increments at 48 km resolution). A prototype of a 4D-Var scheme for AROME-France has
already been tested. On one side, the development of the tangent linear and adjoint versions of the non-linear model is very difficult in a convective scale system. On the other side, the resulting improvement remains weak compared to the numerical cost increase.

At Météo-France, these two limitations are now addressed through the EnVar approach, which consists in specifying background error covariances directly from an ensemble of forecast perturbations, provided by an ensemble of data assimilations
(EDA) (Hamill and Snyder, 2000; Lorenc, 2003; Buehner, 2005). Such fully flow-dependent covariances are then used in a variational context to produce a deterministic analysis. Moreover, as demonstrated in numerous studies (Desroziers et al., 2014; Wang and Lei, 2014; Gustafsson and Bojarova, 2014), this approach makes it possible to derive temporal background error cross correlations, and thus to formulate a 4DEnVar cost function involving a 4D state (i.e. the succession of the different 3D model states along the assimilation window). In this context, the information of an observation at a given time is propagated to
other instants of the 4D assimilation window by the 4D background error covariances, in order to produce an analyzed 4D state. In this way, a 4D scheme can be implemented without developing the tangent linear and adjoint models for complex model components (e.g. micro-physics, non-hydrostatic dynamics, ... ). Furthermore, the numerical cost of this approach is lower than that of 4D-Var. The minimization process does not require any model integration and the numerical cost partly depends on the number of assimilation window timeslots. The main numerical cost increase is due to the computation of the newly required
EDA and can be adjusted with the horizontal resolution or the size of the ensemble for example. The counterpart of EnVar methods is the need for localization, in order to remove long-range correlations in background error covariances affected by sampling noise, due to limited ensemble size.





The development of a 3DEnVar version presented in this study is a first step towards a 4DEnVar scheme for AROME-France, and a major component of the scientific forward planning of Météo-France NWP activities. Such schemes facilitate a lot further

developments, such as the extension of the control variable to hydrometeors (Wang and Wang, 2017; Destouches et al., 2023) for instance, since background error covariances for these new variables are directly available from the EDA. Consequently, the direct assimilation of radar reflectivities or lightning observations (Combarnous et al., 2024) can be considered.

In this context, two early preparatory studies have already been published, regarding preliminary 3DEnVar experiments at Météo-France :

– Montmerle et al. (2018) (MO18 hereafter) presented the formulation and some preliminary tests concerning the localization, hybrid formulation and DA experiment. Results were promising as the 3DEnVar scheme outperformed standard 3D-Var in an experimental study conducted over a 5-week winter period. This study had the advantage to be performed using a prototype of the Object-Oriented Prediction System (OOPS) software framework, which should replace the historical ARPEGE/AROME/IFS software. Nevertheless, it used a non operational configuration with a 3.8 km horizontal

resolution, both for the ensemble and the deterministic model (versus 1.3 km in operations for the latter) and a 3-hour period DA cycle (versus 1-hour in operations).

   – Michel and Brousseau (2021) (MI21 hereafter) evaluated a dual-resolution 3DEnVar scheme (1.3 km for the deterministic model, 3.2 km for the EDA) in an hourly cycle as in the operational system. An hybrid configuration, merging climatological and flow-dependent background error covariances, has been tuned and evaluated over a 3-month summer

period, confirming better performances of hybrid 3DEnVar compared to 3D-Var. Numerous sensitivity studies related to localization lengths, ensemble size or hybrid weights are also depicted. However, this study presents two weaknesses. Firstly, the use of the old historical ARPEGE/AROME/IFS software makes further developments technically much more cumbersome than in the OOPS framework. Likewise, the exclusive use of a hybrid formulation makes some scientific developments much more complex, as the temporal dimension (4DEnVar) or the covariances involving the new variables

(hydrometeors) are not fully managed by the hybrid B matrix, whereas they are in a fully ensemble-based one.

Since the publication of these two studies, numerous investigations have been conducted, in order to be able to use a 3DEnVar scheme in the operational AROME-France system with the following criteria :

   – to tune and implement a robust system in an operational context with its associated constraints : robustness, numerical cost and stability, temporal constraint for the operational use, ...

– to outperform the 3D-Var scheme performances in terms of meteorological quality of the forecast provided to forecasters,

   – to propose a system allowing further developments. This includes the use of the OOPS software, and if possible a full ensemble background error covariance matrix configuration, in order to facilitate the shift towards 4DEnVar planned for the next evolution, or the extension of the control vector to variables missing in the current modeling of the B matrix (e.g. hydrometeors, non hydrostatic variables, surface variables, ...)





The present article aims at summarizing these investigations, and describing the retained configuration and its performances. After the introduction, section 2 will describe the system that is currently in operation. The third section is dedicated to sensitivity experiments performed to tune the system. The performances of the selected configuration compared to the previous 3D-Var scheme will be evaluated using general scores in the fourth section. A focus on different severe meteorological events such as winter storms, fog and High Precipitating Events (HPEs) will be made in section 5, using an original approach to
perform an exhaustive evaluation of the new system.

## 2    Configuration setting

### 2.1    Operational numerical weather system and data flow

The Météo-France Numerical Weather Prediction system used in operation contains two main models and their different versions :

– the global model ARPEGE (Courtier et al., 1994; Bouyssel et al., 2021) using a stretched grid with a horizontal resolution of 5 km over France. It employs a multi-incremental 4D-Var scheme (2 outer-loops) in a 6-hour data assimilation cycle. Background errors are simulated by running a 50-member EDA (Fisher, 2003; Raynaud et al., 2011), allowing flow-dependent covariances to be specified. Since October 2024, the matrix $\mathbf{B}$ is a hybrid combination (Berre and Arbogast, 2024) of covariances localised in gridpoint space and of covariances modeled in wavelet space (Berre et al., 2015), which
are both computed from the EDA.

   – the limited area AROME-France model (Seity et al., 2011) at 1.3 km using a 3DEnVar scheme (instead of the historical 3D-Var which was used before October 2024) in a hourly cycle (Brousseau et al., 2016). An EDA version is run at the lower horizontal resolution of 3.2 km (MI21) in a 3-hour cycle in order to reduce the numerical cost of the 50 members currently used.

One can note that these two main systems are complemented by their Ensemble Prediction System (EPS) versions, ARPEGE-EPS (PEARP, Descamps et al. (2015)) for the global part and AROME-EPS (PEARO, Raynaud and Bouttier (2017)) for the limited area version. Both systems run at the same horizontal resolution as their deterministic versions since the operational upgrade of June 2022. The uncertainties on their initial conditions are sampled by their respective EDA. Different AROME configurations also exist for nowcasting purpose (Auger et al., 2015) and oversea domains (Faure et al., 2020) (without assim-
ilation for the latter). The reader can refer to the different references for more details, only AROME-France data assimilation system and its EDA version (AROME-EDA) are detailed here. The first one uses the 3DEnVar scheme described in this paper in a 1-hour cycle. Its EDA counterpart still relies on the limited area 3D-Var scheme (Fischer et al., 2005) in a 3-hour period cycle, previously used in the deterministic version. They assimilate a wide range of observation types, including radar reflectivities through retrievals of humidity profiles (Caumont et al., 2010; Wattrelot et al., 2014), radial winds from Doppler
weather radars (Montmerle and Faccani, 2009), over Europe (Martet et al., 2022), instruments from various polar orbiting and





geostationary satellites (Guidard et al., 2011) as well as ground-based GNSS atmospheric path delays (Mahfouf et al., 2015) in addition to conventional observations.

The links between the different model implementation are summarized on figure.1. The 1-hour range forecasts of the DA cycle (plain red arrows) use Lateral Boundary Conditions (LBCs) from the latest ARPEGE forecast available, in an asynchronous way : H-5, H-4, H-3, H-2, H-1, H and H+1 analysis times use LBCs from the H-6 ARPEGE analysis time (dotted black arrows on Fig.1). On the contrary, long forecasts useful for operations are launched 8 times a day (every 3 hours) and use synchronous LBCs from ARPEGE, to benefit from the most recent LBCs, with a positive impact on the performances. These long forecasts have also the particularity to be initialized with the AROME analysis valid at H but are also updated thanks to the Incremental Analysis Update (IAU Bloom et al. (1996)) with the H+1 analysis (dashed blue arrows on Fig.1) to benefit from the most recent observations available when this forecast is effectively launched (Brousseau et al., 2016). This particular setup implies that the H+1 analysis has to be performed just after the H one. The reader can note that, in this configuration, the IAU is not used for its filtering properties, but it is employed to update the forecast with more recent observed information.

The operational AROME-EDA aims to simulate the propagation of different errors through the assimilation cycle in order to sample initial uncertainties in the operational AROME-EPS Raynaud and Bouttier (2016)), and to provide background error covariances to the EnVar schemes. Model errors are taken into account by perturbing the total parametrized tendency of physical processes with multiplicative noise (SPPT : Buizza et al. (1999)) implemented in the AROME-EPS by Bouttier et al. (2012). Observation errors are simulated by adding random draws of the observation error covariance matrix. The addition of such model and observation perturbations and their propagation in the assimilation cycle allow analysis and background errors to be simulated (Brousseau et al., 2011). Finally, the LBC errors are represented through the use of perturbed LBCs provided every 3 hours by the global EDA AEARP in an asynchroneous way, regarding the availability of the different forecasts in real time : H and H+3 AROME-EDA assimilation time (where H is either 00, 06, 12 or 18 UTC) use LBCs from the H-6 AEARP assimilation time.

Regarding this organization and the way the different systems are linked and strongly dependent from each other (Fig.1), the 3DEnVar analyses of the new scheme have to use perturbations from AROME-EDA in an asynchronous way : H-1, H, and H+1 analyses use respectively 5, 6 and 7-hour range perturbed forecasts (dashed-dotted green arrows on Fig.1) from the H-6 AROME-EDA analysis time to calculate the perturbations used for the **B** estimation of the hourly data assimilation cycle.

## 2.2 Software framework

Historically, AROME-France has been developed in the same software as ARPEGE and IFS, according to a collaboration between ECMWF, Météo-france and its LAM partners from the ALADIN and HIRLAM consortia (merged now in the AC-CORD consortium). The OOPS project started in 2009 with the objective of renovating the common data assimilation code, in order to enable the development of new algorithms and ease maintenance. It uses an object-oriented design, and consists in adding an upper level code in C++ on the historical IFS-ARPEGE-LAM FORTRAN code, requiring an important refactoring of the latter. The main part of the coding effort is now completed and if different uncompleted prototypes were available in the previous version of the code to begin investigations and developments, the cy48 version is complete and allows to run AROME





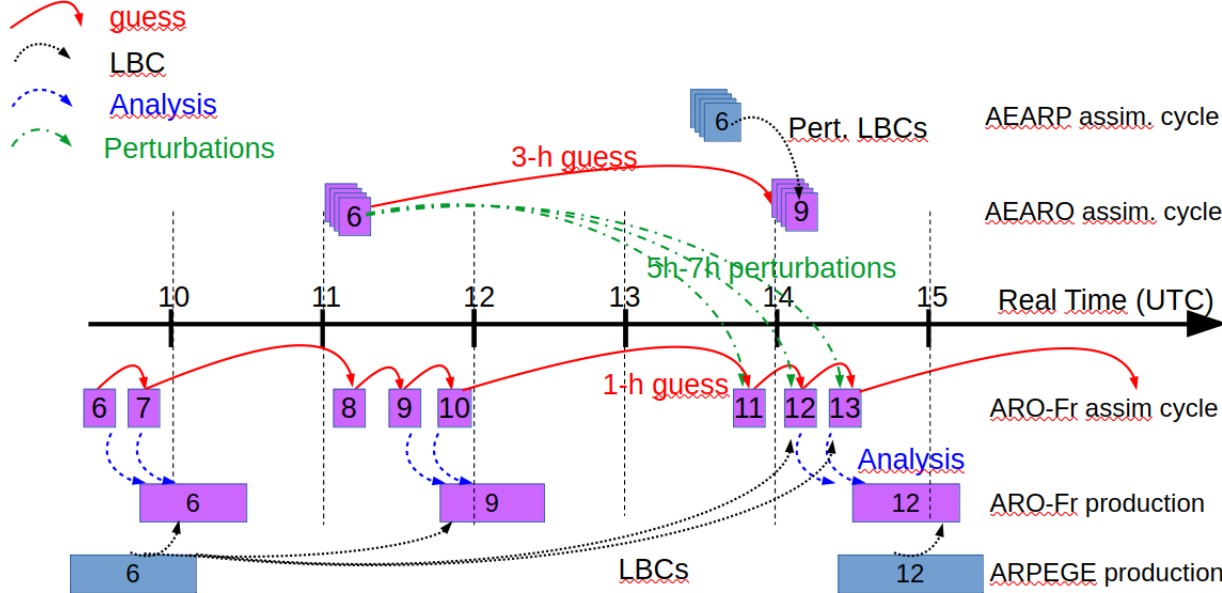

**Figure 1.** Logical scheme of an extract of the temporal implementation of the operational AROME-France model. The rectangles represent the execution of an assimilation-forecast block corresponding to the analysis time indicated by the number in each rectangle. These blocks are located on the time axis accordingly to the real time they are performed. The length of each rectangle is proportional to the wall time used to perform the block on Météo-France HPC.

and ARPEGE DA systems with all the ingredients existing in the historical FORTRAN code (e.g. all operational observation types, variational bias correction, digital filter, ... ). In addition, OOPS allows to use different DA schemes (3D-Var, 3DEnVar, 4DEnVar, hybrid versions, ... ), minimizers or pre-conditioning and to easily modulate the control variable (change or addition of variables). All the experiments used in this study have been performed using the OOPS software (including the 3D-Var reference experiment) in cy46 for the earlier runs or cy48 version for the later ones.

**2.3   3DEnVar formulation**

MO18 and MI21 described the formulations used in both studies to minimize the following incremental 3D cost function $J$ :

$$J(\delta\mathbf{x}) \quad = \quad \frac{1}{2}(\delta\mathbf{x})^T\mathbf{B}^{-1}(\delta\mathbf{x}) + \frac{1}{2}(\mathbf{d} - \mathbf{H}\delta\mathbf{x})^T\mathbf{R}^{-1}(\mathbf{d} - \mathbf{H}\delta\mathbf{x})$$

where $\delta\mathbf{x}$ stands for the analysis increment, the difference between the analysis and the background, $\mathbf{B}$ and $\mathbf{R}$ represent respectively the matrices of background and observation error covariances. $\mathbf{d} = \mathbf{y}^o - \mathcal{H}(\mathbf{x}^b)$ is the innovation vector representing the difference between the observations $\mathbf{y}^o$ and the background $\mathbf{x}^b$ projected in the observation space by the observation operator

$\mathcal{H}$, $\mathbf{H}$ being the linearized version of the latter. This cost function is quadratic and nullifying its gradient leads to solving the



following linear system :

$$(\mathbf{B}^{-1} + \mathbf{H}^T\mathbf{R}^{-1}\mathbf{H})\delta\mathbf{x} \quad = \quad \mathbf{H}^T\mathbf{R}^{-1}\mathbf{d}$$

which needs to be preconditioned for efficiency, through a change of variable. At this stage, different kinds of conditioning are available in the OOPS framework. Among them, the square root form $\delta\mathbf{x} = \mathbf{B}^{1/2}\mathcal{X}$ can be solved with the conjugate gradient algorithm, as historically available in the old ARPEGE/IFS fortran code (used in MI21). Secondly, the full form $\delta\mathbf{x} = \mathbf{B}\mathcal{X}$
can be solved with the Derber-Rosati Preconditioned Conjugate Gradient (DRPCG, Derber and Rosati (1989)) algorithm (also used in MO18). Some elements about advantages and inconveniences of these two pre-conditioning are given in MI21. The 3D-Var scheme involves a climatological parametrized background error matrix $\mathbf{B}_s$. In the AROME-France 3D-Var, $\mathbf{B}_s^{1/2}$ is modelled following the Berre (2000) multivariate formulation, which involves a sequence of linear operators to represent variances, correlations and cross-covariances between the different variables of the control variable. Brousseau et al. (2011)
used an AROME-France EDA at the model resolution to calibrate the different operators. The EnVar scheme is based on the use of a $\mathbf{B}_e$ matrix directly sampled from an ensemble of forecast perturbations, which can also be provided by an EDA, with $\mathbf{B}_e = \mathbf{C}o\mathbf{X}\mathbf{X}^T$, where $\mathbf{C}$ is the localization matrix, "$o$" the Schur product and $\mathbf{X}$, the perturbation matrix where every column contains the deviation of an ensemble member with respect to the ensemble mean divided by $\sqrt{n_e - 1}$ ($n_e$ being the ensemble size). Finally, an hybrid scheme, used in MI21, merges these two approaches with $\mathbf{B} = \beta_s^2\mathbf{B}_s + \beta_e^2\mathbf{B}_e$ where $\beta_s^2$ and $\beta_e^2$ are
scalar weights.

The numerous investigations in MO18 and MI21 lead to define the settings described now. The 3DEnVar scheme uses the $\mathbf{B}^{1/2}$ pre-conditioning and the associated cost function as in MI21, even if the $\mathbf{B}$ pre-conditioning was used in MO18 and also in some experiments of the present study. The localization is done in spectral space as in MO18, without using any recursive filter as in MI21, mainly as a preliminary step towards the use of the Scale Dependent localization (SDL, Caron et al. (2019)).
The same localization is applied to all variables in order to be factorized and applied only once, allowing the numerical cost to be reduced. The control variable includes temperature, surface pressure, specific humidity and the two components of wind u and v, instead of vorticity and divergence, in order to use similar spatial scales when applying this common localization length scale (Berre et al., 2017). Ménétrier et al. (2015) have proposed a theory that allows to diagnose optimal localization length scales using only information from the ensemble. MO18 have used this method to estimate a common localization
length scale in the AROME EDA framework ; they retained and evaluated in the AROME-France 3DEnVar a value of 170 km, constant on the vertical, despite a diagnosed vertical profile increasing with altitude. On their side, MI21 have obtained better performances by reducing this constant value to 40 km. The very important choice of the localization length scales is thus discussed in the following section, in addition to other ingredients, such as the potential use of a hybrid version, and also a coefficient of inflation to adjust the ensemble spread, or the use of an IAU technique to reduce the spin-up.





## 3 Tuning of the 3DEnVar scheme

### 3.1 Methodology

Different parameters have to be tuned in such a 3DEnVar scheme. The authors are aware that all these parameters are linked to each other and that changing one parameter can have an impact on the optimal value of another one. However, the number of experiments that can be run and evaluated is limited : we have to make a choice among all the possible adjustable parameters. Given the preference for a full ensemble **B** matrix, the localization length scale appears to be the most important parameter to investigate first. The other parameters will be examined as a second step. The first goal of these investigations is to build a data assimilation cycle allowing subsequent forecasts to perform better than with 3D-Var. In order to have consistent results, all sensitivity tests will be compared using the diagnostics that have been found to be representative of the obtained results. Two kinds of diagnostics have been selected. The first one is the Root Mean Square (RMS) of observation-guess (O-B) and observation-analysis (O-A) differences in observation space for several observation types. We focused here on observations with a wide spatial and temporal coverage over the AROME-France geographical domain : aircraft measurements of temperature and wind from commercial airlines, that are informative on the larger scales, and radar reflectivities, which are informative on the smaller scales resolved by the system and which are more representative of convective scale phenomena.

The second diagnostic is the precipitation scores resulting from the comparison between i) 24 hr accumulated precipitations observed by the blending product (called ANTILOPE (Laurantin, 2013)) using radar and rain-gauge measurements and ii) 24 hr accumulated precipitation simulated by the sum of the 1 hr precipitation accumulation from the successive 1 hr forecasts of the DA cycle over 24 hr. As the simulated precipitation can be seen as the result of a temporal and vertical integration of the different prognostic fields of the model, this diagnostic appears to be a good measurement of the general behaviour of the system. Relative differences of RMS of O-A and O-B between the different experiments are summarized in figure 2, and relative differences of Heidke Skill Score (HSS) for 24 hr accumulated precipitation in figure 3. The use of 24 hr accumulated precipitation allows a wide range of accumulated precipitation thresholds to be sampled enough to provide statistically significant results.

### 3.2 Horizontal localization length scale

Based on the optimal results of the previous studies, two horizontal length scales constant on the vertical have been first evaluated, namely 150 km as in MO18 (experiment Loc h150 hereafter), and a shorter one as in MI21, corresponding to 25 km (experiment Loc h25 hereafter). O-A and O-B results against 3D-Var (3dv hereafter) are summarized in the two first lines of each panel of Figure 2. For both experiments, RMS(O-A) are strongly smaller in the 3D-Var experiment than in the 3DEnVar experiment, due to lower $\sigma_b$ in the flow-dependent 3DEnVar B matrix than in the climatological modelized 3D-Var one : the analysis fits observations to a lesser extent in 3DEnVar. RMS(O-B) are closer between the two compared experiments. In the Loc h150 configuration, the differences reach only 5% in absolute value, some are degraded (temperature, wind (except at 850 hPa), few are improved. In the Loc h25 experiment, RMS(O-B) values are improved in most of the troposphere but degraded in the high troposphere for both wind and temperature. In order to retain beneficial impacts, while avoiding the degradation in



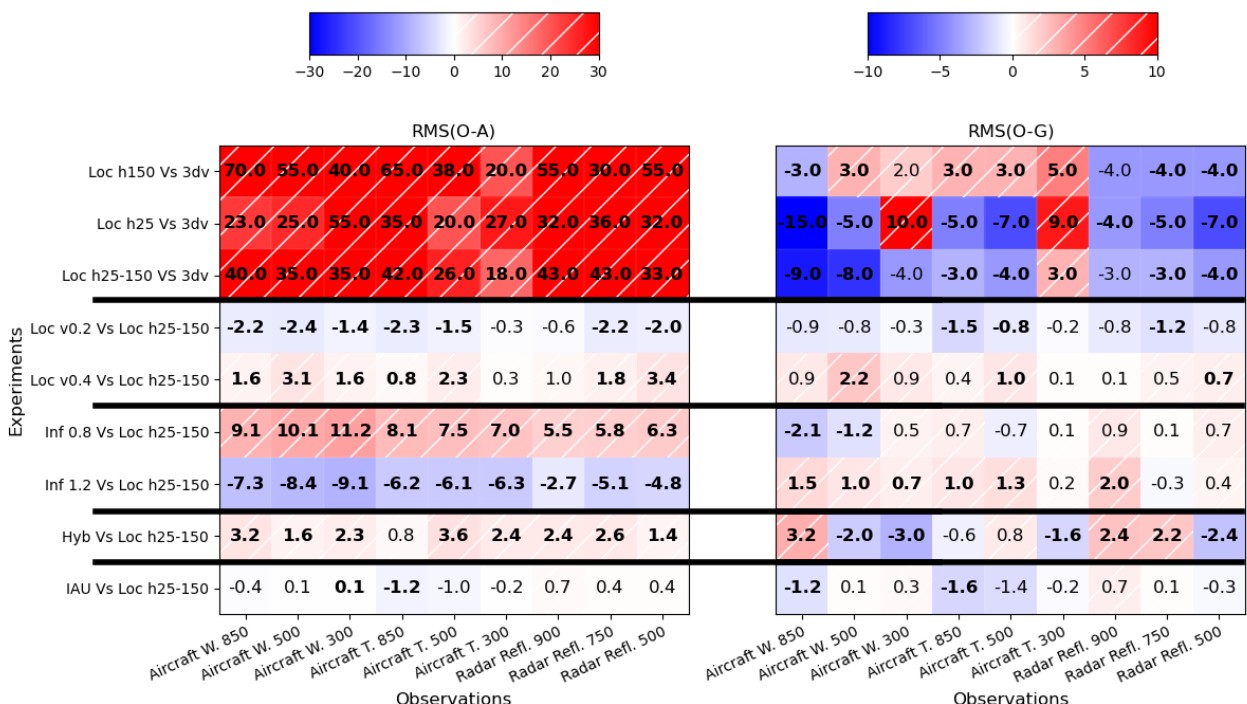

**Figure 2.** Relative difference of RMS(O-A) (left panel) and RMS(O-B) (right panel) between different pairs of experiments (labeled on the left side of each line) for aircraft Wind, aircraft Temperature and radar reflectivity measurements at different atmospheric levels (hPa). Positive (resp. negative) difference values in red (resp. in blue) indicate stronger (resp. smaller) RMS values in the experiment (first part of the left label) compared to the reference (second part of the left label). Bold numbers indicate statistically significant differences using a student test with a 95% confidence interval.

upper levels, an experiment using a horizontal localization length scale regularly varying from 25 km in the lowest vertical level to 150 km in the highest one is performed (experiment Loc h25-150 hereafter). This experiment shows lower improvement

against 3D-Var than Loc h25 in the low troposphere, but remains better than 3D-Var in the high atmosphere. These results are confirmed by the HSS for 24hr precipitation, simulated by the sum of the 24 1-hour range forecasts of the DA cycle for each day (figure 3). The three 3DEnVar experiments show better performances than the 3D-Var one (first three columns of figure 3). Relative HSS differences are higher than 2% and can reach around 6%. They are statistically significant for all experiments and precipitation thresholds. Experiments using the shorter horizontal localization length scales, namely Loc h25 and Loc h25-150,

present the best performances. Indeed, both are better than Loc h150 (fourth and fifth columns of figure 3) and show very close performances : the weak differences between these two experiments (sixth column of figure 3) are not statistically significant.





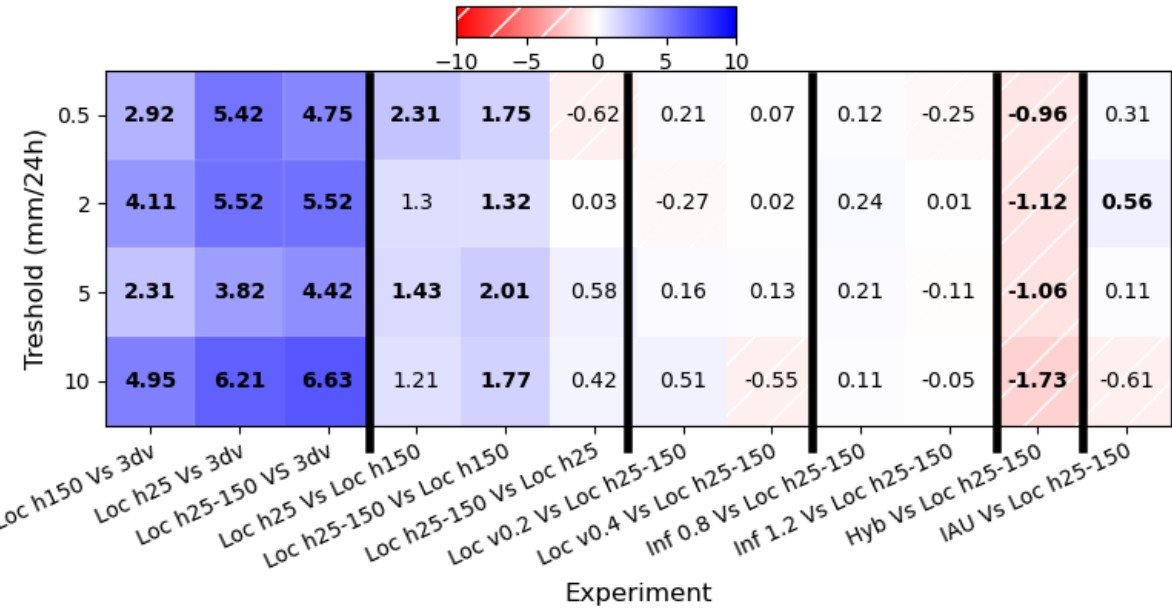

**Figure 3.** Relative difference of HSS between different pairs of experiments (labeled at the bottom of each column). For each experiment comparison, the HSS relative difference for the 0.5, 2, 5 and 10 mm/24 hr thresholds are plotted on corresponding lines. Positive (resp. negative) difference values in blue (resp. in hatched red) indicate improvements (resp. degradations) in the experiment (first part of the bottom label) compared to the reference (second part of the bottom label). Bold numbers indicate statistically significant differences using a boostrap test with a 95% confidence interval.

These findings are different from those of MO18 and MI21 and can be explained by the impact of the localization on the spin-up issue which is discussed now.

### 3.3 Spin-up investigation

Spin-up is referred here as a non realistic propagation of rapid waves in the model, generated by existing imbalances in the model fields. Firstly, it leads, for example, to unrealistic surface pressure tendencies due to gravity wave propagation. In some cases, this phenomenon can also jeopardize the model numerical stability and generate some numerical crashes in case of too important accumulated imbalances. This aspect is investigated by examining the evolution of the RMS of surface pressure tendency averaged over the domain and ten long range forecasts, for the different configurations considered (figure 4). For all

experiments, each forecast starts with high RMS values, which decrease quickly and regularly during the model integration to reach reasonable values after 2 hours of integration. After this forecast range, these diagnostic values are very low : this suggests that these higher averaged values at the beginning of the forecast are due to imbalances in the model states, and are not an indication of meteorological gravity waves, which could be encountered in a non-hydrostatic model solution. This diagnostic also shows that :





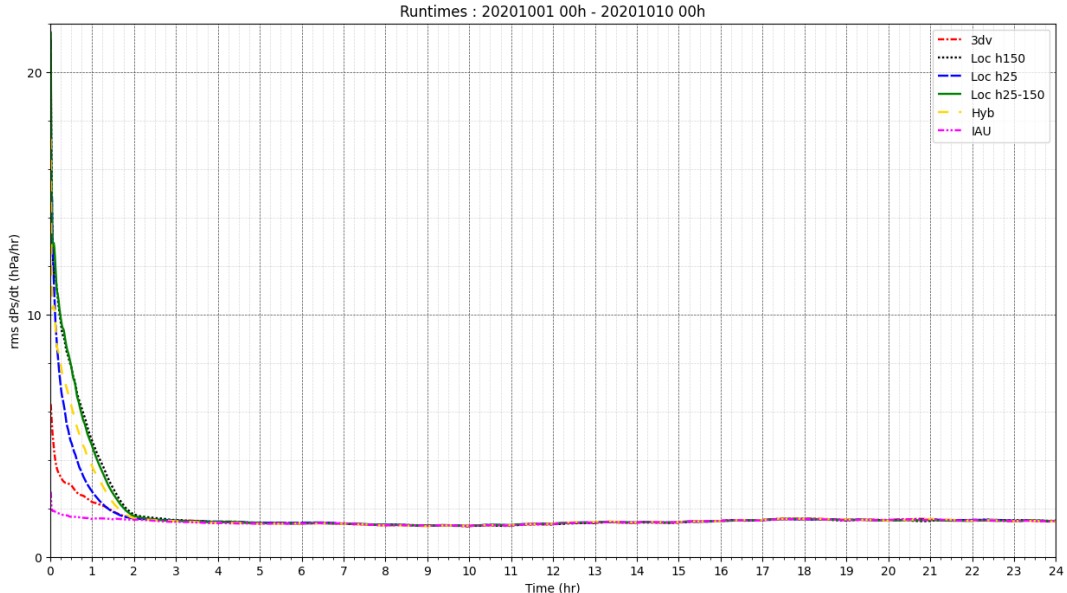

**Figure 4.** Evolution of the RMS of surface pressure tendency (hPa/hr) over the whole domain at each model timestep during the 24-hour forecast ranges averaged over the ten long range forecasts initialized at 00 UTC between 1 and 10 October 2020 for 3dv (dashed-dotted red line), Loc h150 (dotted black line), Loc h25 (short dashed blue line), Loc h25-150 (plain green line) and Hyb (long dashed yellow line) and IAU (dashed-double dotted line) experiments.

1. the spinup is higher in all 3DEnVar experiments not using IAU than in the 3D-Var one,

2. the shorter the localization length scale is, the faster the spin-up decreases during the first hours of forecast (Loc h25 Versus Loc h150),

3. hybridization helps also to reduce the spin-up,

4. the spin-up level is similar for Loc h25-150 and Loc h150 experiments on this diagnostic,

5. as expected, the use of IAU allows to eliminate this spin-up.

The evolution of the RMS of surface pressure tendency averaged over the domain during several successive forecasts of the DA cycle for the different configurations considered has also been investigated (not shown) and it presents a similar behaviour as in the 24-hour forecast. Moreover, while 3DEnVar experiments exhibit higher values at 1-hour forecast range than 3D-Var, these higher values are not accumulated through the DA cycle and remain constant.

Spin-up can also be referred as overestimated precipitation in the first few hours of model integration. This second aspect has already been investigated in section 3.2 and illustrated in figure 3 : whatever the localization length scale chosen, the sum of 1-hour forecasts from the DA cycle of 3DEnVar presents better performances than the 3D-Var one. Therefore, while RMS of





pressure tendency is higher in 3DEnVar, and particularly in the Loc h25-150 experiment, this is not detrimental to the general behaviour of the model and its performances.

Finally, the numerical stability aspect is investigated by comparing the number of numerical crashes in the different experiments. Like 3dv, Loc h25-150 did not crash during the 6 months period investigated, while Loc h150 and Loc h25 experiments crashed both once, during the Alex winter storm on 3 October 2020.

The better results obtained in this study with shorter localization length scales than the optimal ones diagnosed with Ménétrier et al. (2015) could be explained by the fact that, while filtering sampled covariances, the localization helps to filter the numerical noise due to imbalances in the analysis and thus improves performances of the hourly DA cycle used. MO18 used a 3-hour DA cycle and the fast spin-up reduction was not so essential, and MI21 used a hybrid formulation which also helps to reduce the spin-up. Regarding these results, Loc h25-150 shows the best performances on these diagnostics and has been selected as a baseline for the other following sensitivity studies.

### 3.4 Other sensitivity experiments : vertical localization, inflation, hybridization, Incremental Analysis Update

| Name | Horizontal Loc. | Vertical. Loc | Inflation | Hybridization | IAU |
|---|---|---|---|---|---|
| Loc h25-150 | 25-150 km | 0.3 ln(hPa) | 1 | $\beta_s^2=0\ \beta_e=1$ | no |
| Loc v0.2 | 25-150 km | 0.2 ln(hPa) | 1 | $\beta_s^2=0\ \beta_e^2=1$ | no |
| Loc v0.4 | 25-150 km | 0.4 ln(hPa) | 1 | $\beta_s^2=0\ \beta_e^2=1$ | no |
| Inf 0.8 | 25-150 km | 0.3 ln(hPa) | 0.8 | $\beta_s^2=0\ \beta_e^2=1$ | no |
| Inf 1.2 | 25-150 km | 0.3 ln(hPa) | 1.2 | $\beta_s^2=0\ \beta_e^2=1$ | no |
| Hyb | 25-150 km | 0.3 ln(hPa) | 1 | $\beta_s^2=0.2\ \beta_e^2=0.8$ | no |
| IAU | 25-150 km | 0.3 ln(hPa) | 1 | $\beta_s^2=0\ \beta_e^2=1$ | yes |

**Table 1.** Summary of the experiment characteristics used in section 3.4

Different experiments have been performed to evaluate the sensitivity to the vertical localization, inflation (i.e. multiplicative factor on background error standard deviations), hybridization (i.e. use of both ensemble and climatological parts in the B matrix representation) and to the use of IAU to initialize the forecast and reduce the spin-up effects. In this section, the IAU is used in its original form to smooth the imbalances provided by the analysis increment at the beginning of each 1-hour forecast from the DA cycle : the new forecast restarts from the background and a constant fraction (1/36 with a 50s model timestep during 30min) of the new analysis increment (analysis minus background) is added to the model solution at the end of each model timestep during the first 30min of the model integration for the variables of the control vector (u, v, T, q and Ps). The characteristics of the different experiments are summarized in table 1. The results of these experiments are illustrated using the same diagnostics as in section 3.2 : relative difference of RMS of O-A and O-B in figure 2 and precipitation HSS for the sum of the 1 h forecast ranges from the DA cycle in figure 3 compared to the Loc h25-150 experiment. The first result is that the differences obtained on these diagnostics in this section are much smaller than in section 3.2. Relative differences of




RMS(O-B), RMS(O-A) and precipitation HSS can be smaller by one order of magnitude than the changes provided by the switch from a 3D-Var to a 3DEnVar scheme or by the tuning of the horizontal localization length scale studied in the previous part. Differences are also less often statistically significant. In details :

- Reducing (respectively increasing) the vertical localization from 0.3 to 0.2 (resp. 0.4) ln(hPa) can provide some weak improvements (resp. degradations). These results are consistent with MI21 (0.25 ln(hPa)) and MO18 (0.3 ln(hPa)) who also diagnosed an optimal value varying from 0.2 ln(hPa) in low levels to 0.4 ln(hPa) in the mid-stratosphere.

- As expected, reducing (resp. increasing) $\sigma_b$ by a factor of 0.8 (resp. 1.2) leads to increased (resp. reduced) RMS(O-A) as the analyses fit less (resp. more) the observations. The impacts on the subsequent forecasts (the backgrounds) are weak and rarely statistically significant. On these diagnostics, reducing the background error standard deviations, which is a way to compensate for the use of 5, 6 and 7-hour forecast range perturbations from the AROME EDA to estimate background error covariances of the 1-hour cycle, can provide some slight improvements. Nervertheless, these positive impacts are not propagated to the longer forecast ranges, and lead to deteriorated performances in the lowest model layers (not shown). A stronger $\sigma_b$ reduction (inflation factor of 0.5) confirms these worse results (not shown).

- The use of the hybridization ($\beta_e = 0.8$ and $\beta_s = 0.2$ is illustrated here) can help to slightly improve some O-B values. Nevertheless, It leads to statistically significant weaker performances on the HSS of precipitation simulated by the sum of the 1-hour range forecasts (figure 3). Scores on longer forecast ranges have also been investigated (not shown) and present the same kind of degradation.

- The use of the IAU seems to provide some slight improvements, but once again rarely statistically significant.

Finally, different combinations of values and mixture of the previous components have also been evaluated but the impact were not interesting and are not discussed here. On the other side, the results of MI21 concerning the better performances provided by a 50 member ensemble versus a 25 member ensemble have also been confirmed (not shown).

## 3.5 Numerical cost

As in MI21, some adjustments had to be done in order to keep the numerical cost of the new scheme compatible with an operational use. The 3D-Var scheme needs a 260 s wall time to perform the analysis with 50 iterations (inner loop) on 8 nodes of the "taranis" Bull Sequana XH2000 platform from Météo-France equipped with AMD Rome technology (AMD EPYC 7742 64C 2.25GHz). These 8 nodes offer sufficient memory to allow the reading of the 50 ensemble members in the 3DEnVar scheme. However, the additional time to read these 50 members, estimate the perturbations and mainly apply the localization can increase the wall time up to 600s. In order to compensate for this increase, the maximum number of iterations in the inner loop is reduced from 50 to 40 without any impact on the analysis quality (not shown). On the other side, the number of used nodes is increased from 8 to 12 (among the 1152 available) without any impact on the calculator exploitation and side effects on the others components of the NWP system.





## 3.6 Summary

Using the two diagnostics described above, an "optimal" configuration, hereafter called 3dev experiment has been chosen using :

- perturbations from a 50-member AROME-France EDA at 3.2km resolution,

- a varying horizontal localization length scale on the vertical from 25 km at the lowest level to 150km at the highest one,

- a constant vertical localization of 0.3ln(hPa),

- an inflation factor of 1.0 : the dispersion of the 5, 6 an 7-hour perturbed forecast ranges from the AROME-EDA is considered to be realistic and still representative of background errors,

- no hybridization.

This configuration has been investigated over a long time period and on the simulation of different meteorological severe events such as winter storms, fog or HPEs. The correct numerical stability of the model dynamic in this configuration has been demonstrated by the fact that no numerical explosion has been reported during more than 1 year of experiments despite the simulation of numerous storms with strong winds. Surprisingly, this configuration met numerical problems while simulating

heat-waves during the 2022 summer : four 1-hour forecasts from the assimilation cycle crashed at the beginning of the model integration in mountainous areas, while the temperature analysis increment provided by surface stations was concentrated near the ground by the flow-dependent background error vertical correlations from the 3DEnVar scheme compared to the climatological 3D-Var ones.

As a precaution, a second configuration has also been investigated in parallel. It is called 3devi in this study, and it uses IAU

to smoothly introduce the analysis increment in the first 30 minutes of the model forecasts in the assimilation cycle, and also in the longer range forecasts.

## 4 Evaluation of the selected configuration over a long time period

This section presents the results of the comparison between the 3D-Var (3dv) and 3DEnVar experiments (3dev and 3devi) retained after the previous section over a 5.5 months period using different scores.

## 4.1 Usual scores

First, classical scores estimated against the ECMWF/IFS analysis, radio-sounding and surface station measurements are investigated. The relative differences of the Root Mean Square Error between 3dev and 3dv experiments for different model variables, levels and forecast ranges are summarized in figure 5. Concerning the tropospheric fields, 3dev is generally closer to the reference than the 3dv experiment. Rare exceptions concern geopotential (IFS analysis and radio-sounding) and temper-

345 ature (only for radio-sounding) at the analysis time, as already seen for O-A values in figure 2. This RMSE reduction in 3dev





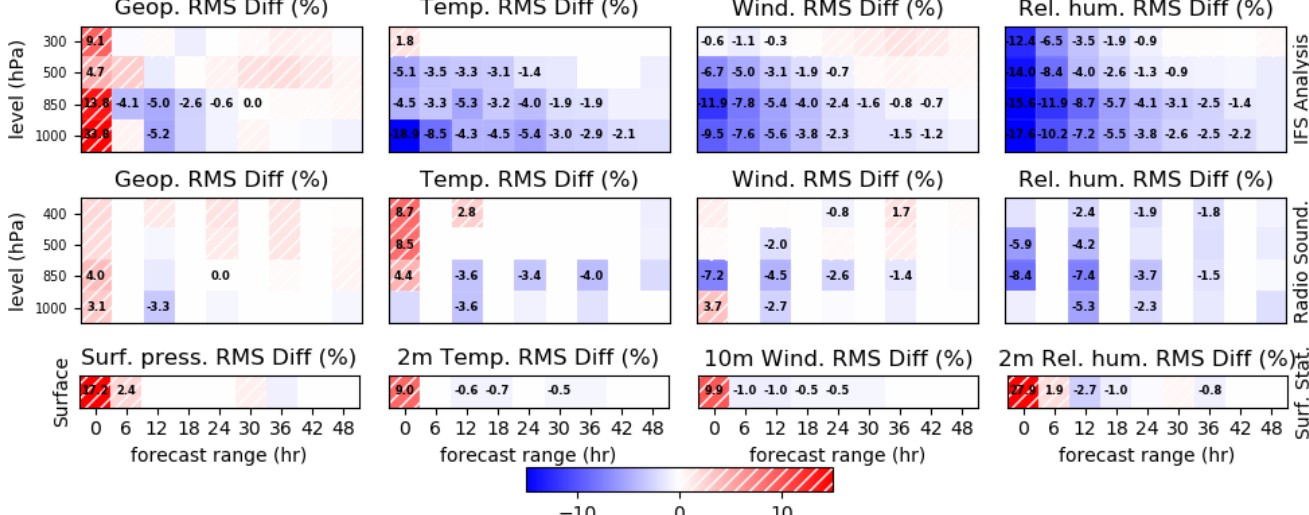

**Figure 5.** ScoreCard of the relative difference of Root Mean Square Error between 3dv and 3dev experiments averaged from 15 of September 2020 to 1st of March 2021, for geopotential and surface pressure (first column), temperature (second column), wind (third column) and specific or relative humidity (last column) against IFS analysis (first line) and radio-sounding (second line) at different vertical levels (y axes), and surface stations (last line) from 0 to 48 h forecast ranges (x axes) every 6 h (12 h for radio-sounding). Negative values (in blue) indicate better performances for the 3dev experiment. Bold numbers indicate that the difference is statistically significant using a Bootstrap test with 95% confidence interval.

experiment is higher for lower levels and earlier forecast ranges and decreases when forecast range increases, as the impact of the initial conditions is reduced in long forecast ranges. Using IFS analysis as reference, these differences still reach 5% for all the fields at 12-hour forecast range and 4% at 24-hour forecast range for temperature and specific humidity. Most of these differences are statistically significant (as indicated by the presence of a number in the corresponding boxes), particularly for

temperature, wind and specific humidity. Regarding radiosondes, the improvements are weaker but still significant for almost the same forecast range albeit over fewer vertical levels. Concerning surface observations, the behaviour of 3dev is close to 3dv except at analysis time.

The use of IAU in the 3devi experiment, compared to 3dev in figure 6, improves the analysis fit to observations. For the other forecast ranges, differences between these two experiments are very much smaller than those with 3dv experiment and

rarely reach 2%.

## 4.2 Precipitation and wind gust probabilistic scores

This first general evaluation is completed using probabilistic scores of precipitation and wind gusts which are more adapted to verification at convective scale, as they take into account the double penalty on the prediction of small scale events. The Brier Skill Score (BSS, Amodei et al. (2015)) of an event (in general a threshold excedence) is estimated comparing the frequency



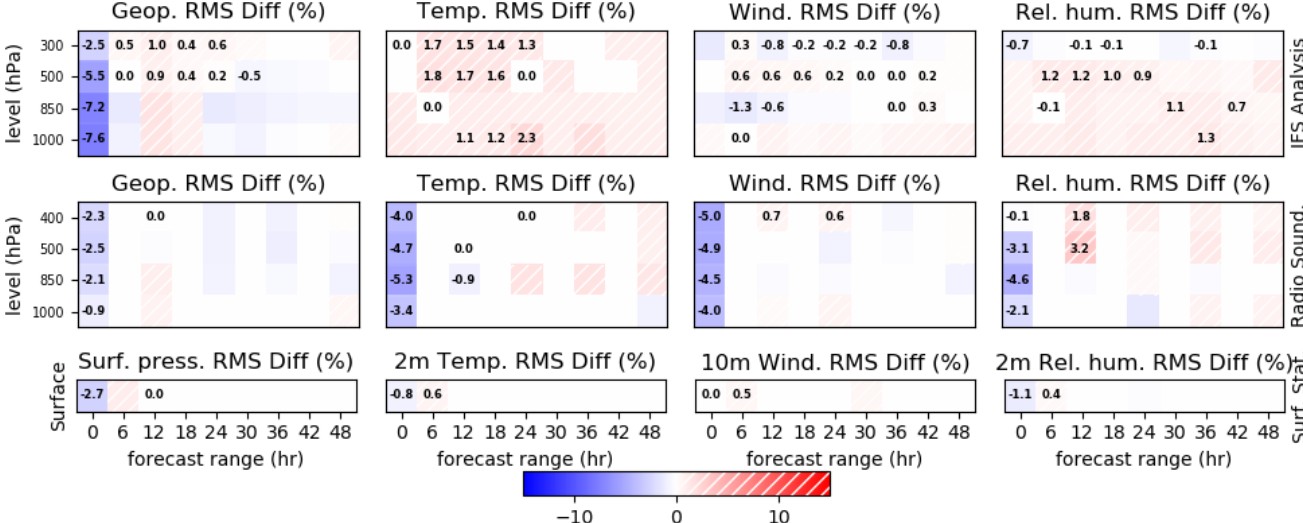

**Figure 6.** Same legend as figure 5 but between 3dev and 3devi experiments : Negative values (in blue) indicate better performances for the 3devi experiment

of the observation and the simulation of this event in a spatial neighbourhood. The relative difference of BSS between 3dv and 3dev experiments are shown in figure 7.a) for the convective period from 15 of September to 1 of December 2020 and figure 7.b) for the winter period from 1 December 2020 to 1 of March 2021 without neighbourhood (in this case BSS is equivalent to HSS) and with a 52.8km spatial neighbourhood. The latter is used to estimate the performance index employed to measure the evolution of the AROME-France performances averaging values of 0.5, 2.0, 5.0 mm/6hr precipitations and 40km/h wind gust BSS for the 6, 12, 18 and 24-hour forecast ranges. As expected, the BSS comparison between the two periods shows that the 3DEnVar analysis has a more important impact during the convective period, as initial conditions have a larger influence on the simulation than during the winter period. This impact is visible for all precipitation thresholds. It is weaker when using the 52.8km neighbourhood, but it remains statistically significant and higher than 4% for the 18-hour forecast range. For wind gust, the improvement concerns the lower threshold and is weaker and less significant than for precipitation. However, the improvement provided by the use of a 3DEnVar analysis in the selected configuration is obvious and visible on the AROME-France Performance Index which changes from 0.814 to 0.823 (+1.1%) during the winter period and from 0.800 to 0.822 (+2.75%) during the convective one which is consistent with the findings of MI21. Using these scores, performances of 3devi are very close to those of 3dev and are not shown and discussed here.

As shown in this section, the retained configuration exhibits better general performances than the operational 3D-Var. It is now evaluated on its ability to simulate particular meteorological events : fog and storm cases during winter, and Mediterranean High Precipitating Events during autumn.



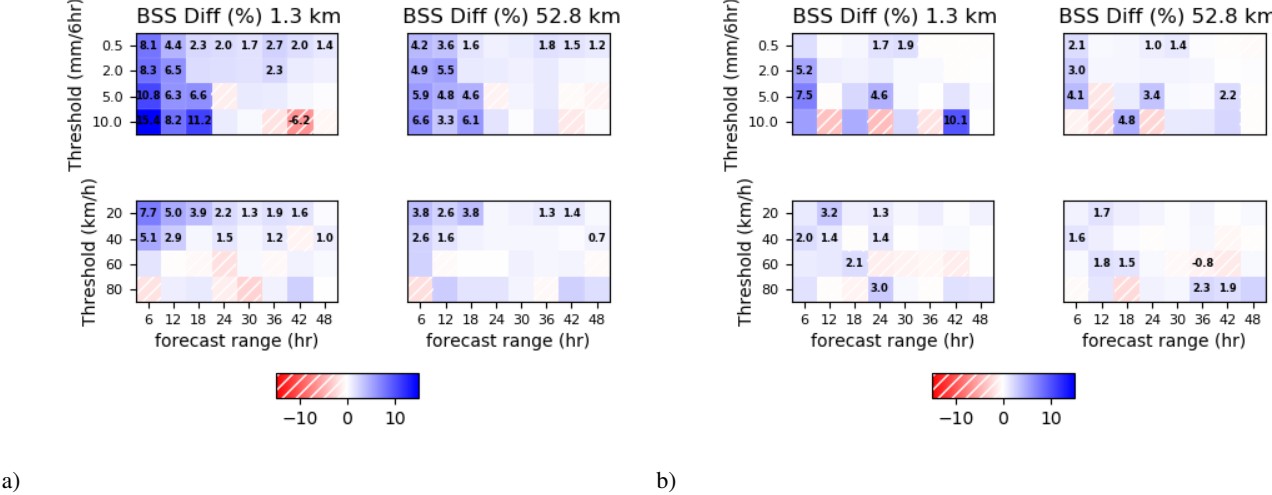

a)                                                                                          b)

**Figure 7.** ScoreCard of the relative difference of Brier Skill Score between 3dv and 3dev experiments averaged from 15 of September 2020 to 1 of december 2020 (a) and from 1 of december 2020 to 1 of March 2021 (b) for 6 h precipitations against ANTILOPE radar/raingauge blending product (first line) and 6 h maximal wind-gust against surface stations (second line) for different threshold (y axes : 0.5, 2, 5 and 10 mm/6h for precipitation, 5.6, 11.1, 16.7, 22.2 m/s) from 0 to 48 h forecast ranges (x axes) every 6 h estimated without (first column) and with a 52.8km spatial neighbourhood (second column). Positive values (blue) indicates better performances for the 3dev experiment. Bold numbers indicate that the concerned difference is statistically significant using a Bootstrap test with 95% confidence interval.

# 5   Evaluation on severe meteorological situations

## 5.1   Winter storms

The performance of the 3DEnVar configuration on the prediction of storms and particularly on the intensity of wind gusts
was evaluated on 6 winter storms over French territory and compared to the 3D-Var one. For each storm, 48-hour forecasts are computed every hour starting from an analysis from the hourly DA cycle. A large sample of long-term forecasts is thus available over the period of the event. Contingency table scores for 80 and 100 km/h wind gust thresholds are calculated using French surface station measurements for the different forecast ranges (1, 2, 3, 6, 9, 12, 15, 18, 24, 30 and 36 hr) which have a validity time during the storm period. The following storms are studied : Daniel, from 16 December 00UTC to 17 December
2019 12 UTC (36 forecasts), Fabien, from 21 December 12UTC to 22 December 2019 18UTC (30 forecasts), Jorge, from 27 February 00 UTC to 28 February 2020 00 UTC (24 forecasts), Alex, from 1 October 00 UTC to 2 October 2020 00 UTC (24 forecasts), Barbara, from 20 October 12 UTC to 21 October 2020 18 UTC (30 forecasts) and Bella, from 27 December 00 UTC to 29 December 2020 00 UTC (48 forecasts).

For instance, for the Daniel storm, the 1-hour HSS is estimated from the 1-hour forecast range from each of the 36 forecasts
starting between 15 December 23 UTC and 17 december 11 UTC, the 2-hour HSS is estimated from the 2-hour forecast ranges from the 36 forecasts starting between 12 December 22 UTC and 17 December 10 UTC ... and the 36-hour HSS, is estimated



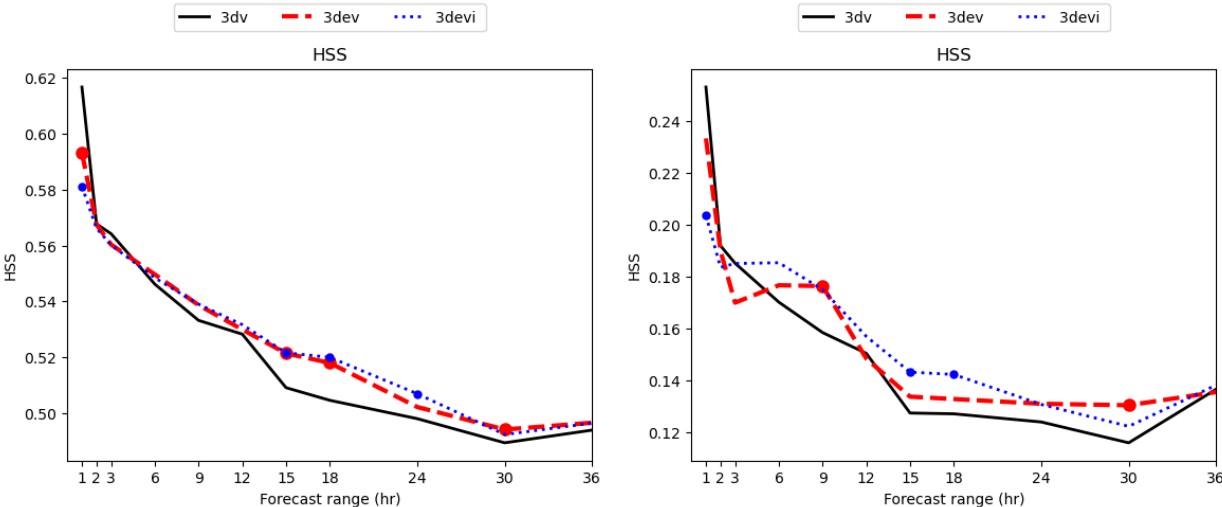

**Figure 8.** HSS for 80km/h (left) and 100km/h (right) wind gusts simulated during 6 winter storms over the french territory (Daniel, Fabien, Jorge, Alex, Barbara and Bella) by hourly initialized forecasts as a function of the forecast range (hr) for 3dv (plain black line), 3dev (dashed red line) and 3devi (dotted blue line) compared to surface station measurements. Colored dot above the lines indicates that the HSS difference with the 3dv experiment is statistically significant for the indicated forecast range using a bootstrap test with a 95% confidence interval.

from the 36-hour forecast ranges from the 36 forecasts starting between 14 December 12 UTC and 16 December 00 UTC. 192 different forecasts are thus used to estimate the Frequency Bias and the HSS presented on figure 8. For both thresholds and all forecast lead times, the two experiments are very close in average. There are few differences over the first forecast ranges, for 395 which 3dv performs better, possibly due to dynamical fields being affected by a higher and longer spin-up in 3dev and 3devi. Fortunately, this longer spin-up does not have negative impact on longer forecast ranges and 3dev can be slightly better (at 15 or 18 h for 80km/h or 9 h for 100km/h). On average, the three experiments seem to present similar performances. This result is not surprising as this kind of meteorological event is driven by large scales and limited area analysis at smaller scales are not expected to be decisive. However, the fact that the smaller scales analyzed with 3DEnVar, and the associated spin-up increase, 400 do not degrade the wind gust simulations, is a very interesting result. The use of IAU in 3devi provides similar results except at 1-hour forecast range which shows a higher degradation than 3dev. At the other forecast ranges, the differences with 3dv remain very close, even if the statistical significance of these differences can differ from an experiment to another.

## 5.2 Fog events

Unlike winter storms, fog events are very sensitive to very local conditions. Therefore, the improvement of their simulations 405 can be expected when using a 3DEnVar analysis, e.g. through vertical error correlations in lower layers, which are more representative of temperature inversions involved in this kind of situations. The ability to simulate fog events has been evaluated in the framework of the SOFOG3D field campaign (Burnet et al., 2020), which took place in the south-west of France during

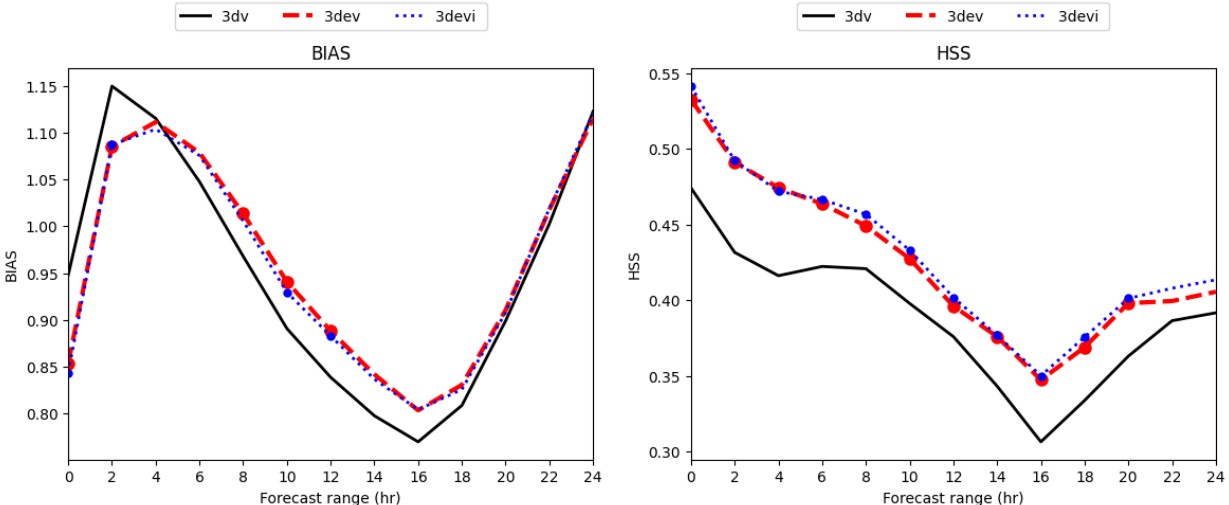

**Figure 9.** Frequency Bias (left) and HSS (right) for fog events (visibility lower than 1000m) simulated by forecasts initialized at 18, 20, 22, 00, 02, 04 and 06 UTC during 16 nights of SOFOG3D field campaign IOPs as a fonction of the forecast range (hr) for 3dv (plain black line), 3dev (dashed red line) and 3devi (dotted blue line) experiments compared to surface station measurements. Colored dots above the lines indicate that the score difference with 3dv is statistically significant for the indicated forecast range using a bootstrap test with a 95% confidence interval.

winter 2019-2020. 48-hour forecasts have been performed starting at 18, 20, 22, 00, 02, 04 and 06 UTC during the nights when stable winter conditions, favourable to fog formation, were expected. In addition, Intense Observation Periods (IOPs)
were planned during the SOFOG3D field campaign, whether the fog was observed or not, in order to measure detections, non detections but also false alarms. 3D-Var and 3DEnVar experiments have been evaluated by computing contingency tables of fog occurrence. Observations came from the French surface stations (a visibility lower than 1000m being considered as a fog event) and the simulations were deduced from a new visibility diagnostic calculated using a second degree polynomial relation that links six months of observed visibility to AROME-France forecasts of liquid water and rain contents (Ingrid Etchevers
personal communication). Frequency Bias (closer to one the frenquency bias value, better the forecast performance) and HSS (higher the HSS value, better the forecast performance) for the event "visibility lower than 1000m" are presented on figure 9. Both 3DEnVar experiments (3dev and 3devi) perform better than 3dv for both scores and the HSS differences are statistically significant for the first 14-16-hour forecast ranges.

### 5.3 High Precipitating Events over Mediterranean area

The improvement of the simulation of High Precipitating Events (HPE) is one of the main goals of a convective scale DA system as AROME-France. This kind of meteorological event is a very important issue for weather forecasting in the south of France as they lead to very important damage and numerous casualties. They occur regularly over these French regions in





autumn when low geopotential areas over the near Atlantic or the Iberic peninsula generate low layer southerly flows which bring hot and moist air masses from the Mediterranean sea. The latter generate strong convective events when they meet the first hilly areas. The main difficulty is to evaluate if this strong convection will remain on mountainous areas or will give stationary precipitations over more populated areas near the Mediterranean seaboard. A 3DEnVar configuration has been first evaluated on 7 HPEs during autumns 2020 and 2021. The behaviour of the data assimilation cycle itself is first investigated by comparing the observed rainfall and the sum of the 1-hour forecasts from the DA cycle, as mentioned in section 3.1. As a second step, long-term forecasts are investigated. As the predictability of such events is very low, it is not sufficient to compare one or two forecasts from different experiments to be able to conclude on their respective performances. In this study, each event is investigated using a lagged ensemble approach : the simulation of the HPE is repeated 16 times for each experiment, using long-term forecasts initialized with successive hourly analyses from the DA cycle. The experiments are evaluated using the full sample in order to obtain a robust comparison.

### 5.3.1    The 19 September 2020 case in detail

The first event investigated occurred on 19 September 2020 in the Gard region and gave more than 500 mm in 20 hours over the same limited mountainous area. Damages were not very severe as this is not an urban area, but important cities are not very far and a slight difference on the location of the maximum of precipitation, as simulated by some of the operational AROME-France runs, would have had a higher impact. The uncertainty on the location of this huge amount of precipitation is an important issue for forecasters in terms of forecasting and warning.

First, the total rainfall observed by the blending product ANTILOPE (based on radar and rain gauges) between September 19 at 04 UTC and September 20, 2020 at 00 UTC (figure 10.a) is compared to the sum of the 1-hour range forecasts from the DA cycle (figure 10.b, c and d). Both 3D-Var and 3DEnVar experiments are able to forecast the correct location of the maxima, but its intensity is underestimated and reaches only 228 mm in the 3D-Var experiment and 253 mm in the 3dev one, far from the 500 mm observed in the blending product or the 375 mm actually measured by the rain gauge of Le Vigan station (figure 11). On the other side, the 3devi experiment reaches 325 mm. This result shows that, despite a higher spin-up, 3DEnVar is able to simulate the phenomenon, even in the shorter lead times, as well as 3D-Var. Longer forecasts from the lagged-ensemble are investigated using ensemble diagnostics. The ninetieth percentile of precipitation is plotted and compared to ANTILOPE product on figure 10.e, f and g). While the 3D-Var experiment exhibits 2 maxima of precipitation, the successive 3DEnVar simulations are more consistent with each other and mainly target the observed location. However, the intensity of the maxima remains close in the three experiments and differences are not statistically significant. To go further, the location of the maximum of precipitation for each forecast of the lagged ensemble built from the 3dv and 3devi experiments is plotted on figure 11.a) ; moreover, the temporal evolution of these maxima during each forecast is plotted also for both experiments using box plot on figure 11.b), in order to compare the simulated intensities and their dispersion. For each lagged-member the location error (the distance between the observed and simulated maximum position) and the intensity error (difference between the observed and simulated maximum intensity) is estimated and their mean and standard deviation for each experiment are indicated on figure 11.a). The results confirm the general better accuracy of the 3devi experiment location of the event. Mean



and standard deviation of the location error reach respectively 33 and 19 km in the 3dv experiment and are reduced to 24 and 9 km in the 3devi one. 3devi exhibits also higher intensities (the size of the triangle indicating the position of the maximum is proportional its the intensity). The intensity error mean and standard deviation are also reduced from 182 and 52 mm in 3dv to 120 and 37 mm in 3devi. The lagged-ensemble approach makes it also possible to evaluate the ability of the system to converge toward a realistic solution while the data assimilation cycle progresses. On figure 11.a), the color shade of the triangle depends on the analysis time used to start the forecast : the darker the shade, the more recent the analysis. Thus, the general tendencies along the different analysis times of the location and intensity errors can be estimated. Both experiments exhibit negative tendencies of intensity error : the more recent lagged-members simulate higher intensity, as also illustrated by the larger size of the darker (more recent) triangles. The location error tendency is also slightly negative for the 3DEnVar experiment and actually, the more recent maxima are closer to the observed location than the older ones. This is not the case for the 3D-Var experiment : the general location error tendency is positive and important and recent maxima can be located far from the observed position. Figure 11.b) confirms that 3devi is able to simulate higher intensities and maximum rainfall. Moreover, the temporal evolution shows that the higher values for 3dv are obtained thanks to the simulation of a second precipitating period during the second part of the event which is not observed, according to the rain-gauge measurements at Le Vigan (black dotted line) and less visible in the 3devi experiment.

### 5.3.2 Summary of the other cases

Similar studies, using the same tools, have been performed on six other HPEs. The results of the comparison between 3dv and 3devi are illustrated on figure 12, except for the event which occured on October 2, 2020.

- October 2, 2020 (not shown) : The Alex storm generated a HPE over the Alpes-Maritimes area which caused very important damage and casualties (Vésubie valley disaster). Unlike the September 19 case, this event is driven by large scale conditions and its predictability is very high. The operational 3D-Var AROME-France system was very efficient and forecasters could trigger an appropriate warning. 3DEnVar experiments perform as well as the 3D-Var one, both on the location and the intensity of the event.

- September 9, 2021 (figures 12.a) and 12.b)) : this event affected two different areas. The 3D-Var operational system encountered two difficulties : the non detection of a stationary thunderstorm over the town of Agen and a false alarm due to the simulation of a HPE over the Aude region which was not observed (only 76 mm were measured), knowing that this area had been strongly affected by very important flash-flooding during October 2018. For both issues, 3devi performs better than 3dv : the stationary cell over Agen is, in average, simulated with better location and intensity. Over the Aude area, the intensity is less overestimated.

- September 14, September 25, October 3, 2021 : These 3 events affected the Gard region. The stakes were the same as for the September 19, 2020 event : the issue is to correctly estimate the location and the intensity of the maximum rainfall over the plains (September 14, figure 12.c) or the mountainous and less urbanised areas (September 25, figure 12.d and October 3, figure 12.e). The higher dispersion of the ensemble for the September 25 event indicates a higher uncertainty





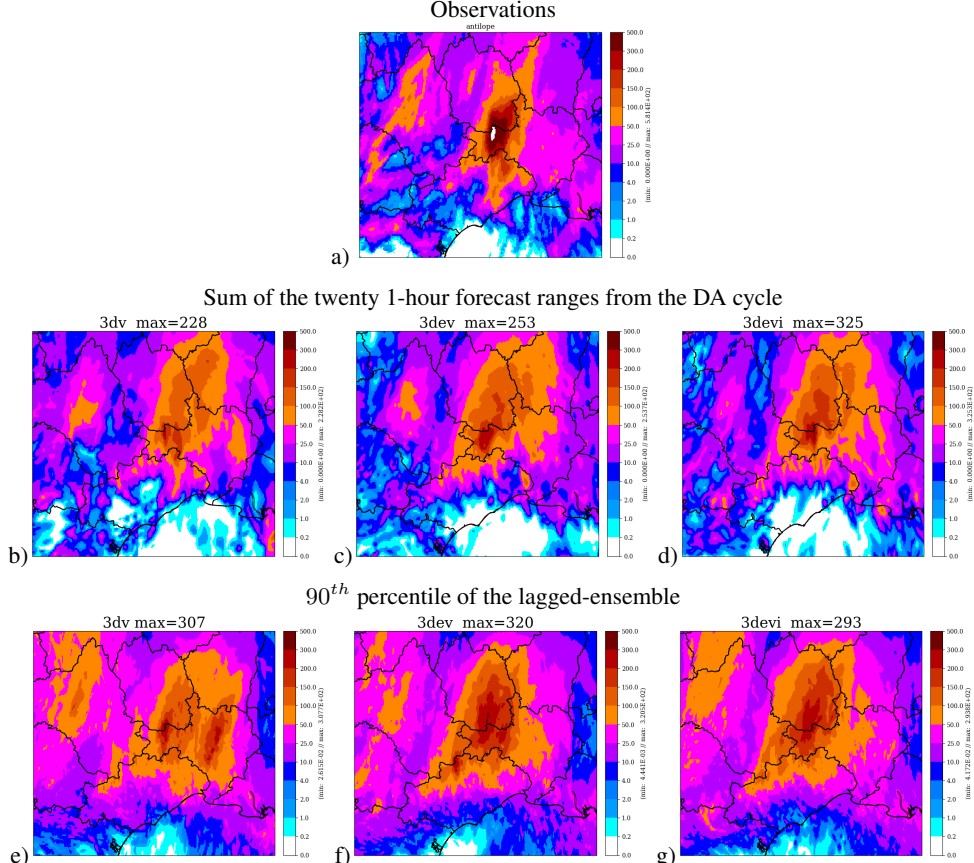

**Figure 10.** Total rainfall between September 19, 2020 at 04 UTC and September 20, 2020 at 00 UTC measured by the blending product (radar and rain-gauges) ANTILOPE (a) and simulated by the sum of the twenty 1-hour forecast ranges from the DA cycle (second line) and the ninetieth percentile of the lagged-ensemble built with the sixteen long-term forecasts hourly initialized from September 18, 2020 at 13 UTC to September 19, 2020 at 04 UTC (third line) for 3dv (left), 3dev (middle) and 3devi (right).

than for the two others events. On this aspect, the 3dv and 3devi experiments have equivalent performances, but 3devi presents a better location and the reduction of the error location along the analysis time is better for these three events (darker triangles are closer to the observed maxima in 3devi than in 3dv).

– October 4, 2021 (figures 12.f): the weather situation encountered the day before continued, with important rainfall over the Var region, which was not correctly simulated by the 3D-Var operational system. 3DEnVar (3devi) helps to simulate 495      more realistic intensities and to target the hinterland areas, even if it is sometimes too eastward.

All the cases studied suggest that the retained 3devi configuration generally makes it possible to improve the simulation of HPEs compared with the 3dv configuration, even if this is not systematic. More accurate analyses that are more representative



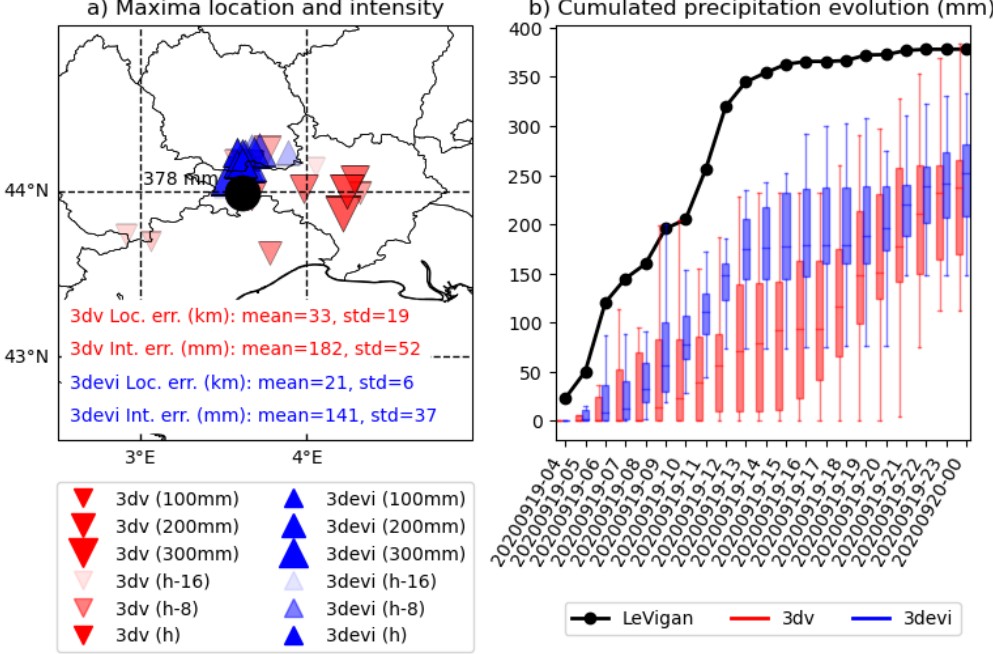

**Figure 11.** 19 September 2020 event : maxima location and intensity for each member of the lagged-ensemble (a) and temporal evolution of the maxima intensity during the forecast (b) measured by Le Vigan rain gauge (black dot) and simulated from September 19, at 04 UTC to September 20, at 00 UTC by each member of the lagged-ensemble by 3dv (red bottom triangle/boxplot) and 3devi (blue up triangle/boxplot) experiments using forecasts initialized from 18 September 13 UTC (H-16) to 19 September 04 UTC (H).

of the small scales involved in such phenomena explain these results. The results of the 3dev experiment are not shown here but are very close to those of 3devi.

### 5.3.3 Impact on the system jumpiness

The jumpiness refers here to the run-to-run consistency and the fact that, for the same event, the different successive forecasts produced by the same NWP system can differ more or less, depending on the predictability of the event. Investigations about these seven HPE cases indicate that 3DEnVar can help to reduce the jumpiness (i.e. improve the run-to-run consistency) of the AROME-France system. The forecasters made subjectively the same analysis, when comparing the AROME-France operational (3D-Var) and its E-suite using 3DEnVar, during the period where both sets of runs were available on other HPE cases. In order to objectively measure this improvement, correlations between the accumulated precipitation fields, respectively observed by the ANTILOPE product and simulated by the different hourly initialized forecasts (from H, the more recent, to H-15, the oldest, columns), both valid at the same time, have been computed for 3dv (a) and 3devi (b) experiments. The results have been averaged on the 8 HPE described in this section, and they are plotted on figure 13. The first column corresponds to correlations between the observation fields and the different lagged forecast fields : it can be noticed that the correlation





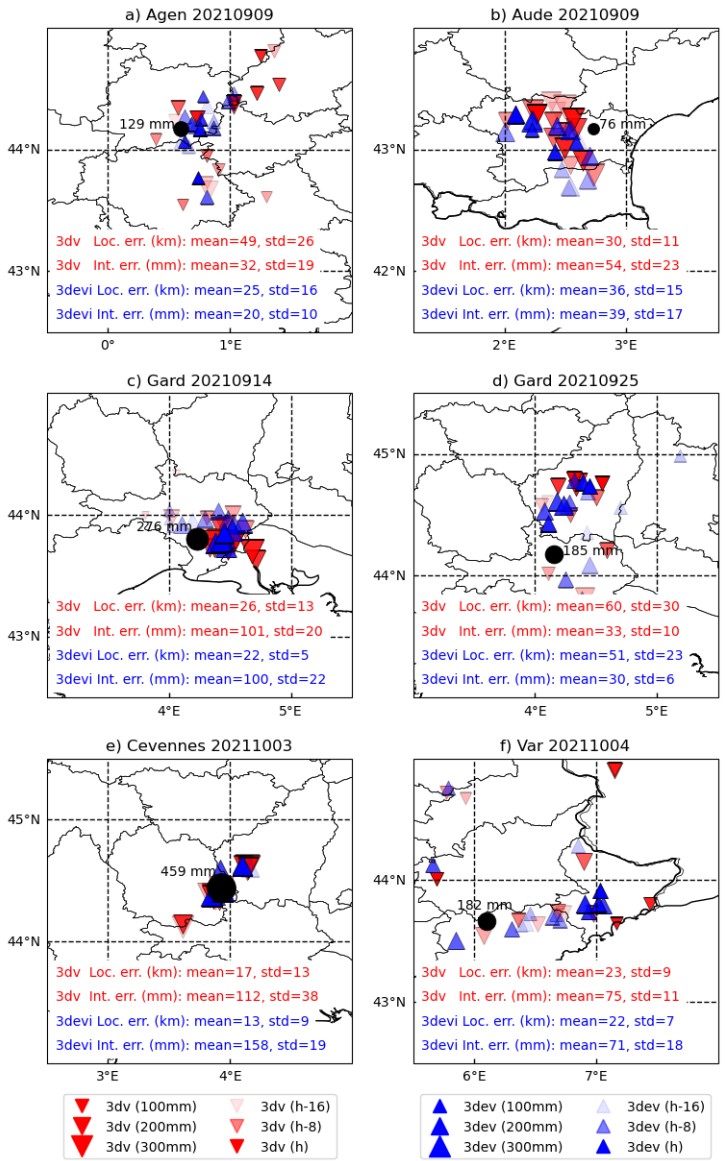

**Figure 12.** Same legend as figure 11.a) but for HPE observed and simulated on different areas for a) and b) from the 08 Sep. 2021 at 18 UTC to the 09 Sep. 2021 at 06 UTC by the 16 forecasts hourly initialized from the 08 September 04 UTC (h-16) to the 08 Sep. 18 UTC (h), for c) the 14 Sep. 2021 from 06 UTC to 12 UTC by the 16 forecasts hourly initialized from the 13 Sep. 2021 at 15 UTC (h-16) to the 14 Sep. 2021 at 06 UTC (h), for d) from the 25 Sep. 2021 at 12 UTC to the 26 Sep. 2021 at 12 UTC by the 16 forecasts hourly initialized from the 24 Sep. 2021 at 22 UTC (h-16) to the 25 Sep. 2021 at 12 UTC (h), for e) from the 3 Oct. 2021 at 00 UTC to the 4 Oct. 2021 at 00 UTC by the 16 forecasts hourly initialized from the 2 Oct. 2021 at 10 UTC (h-16) to the 3 Oct. 2021 at 00 UTC (h), and for f) from the 4 Oct. 2021 at 06 UTC to the 5 Oct. 2021 at 06 UTC by the 16 forecasts hourly initialized from the 3 Oct. 2021 at 16 UTC (h-16) to the 4 Oct. 2021 at 06 UTC (h).



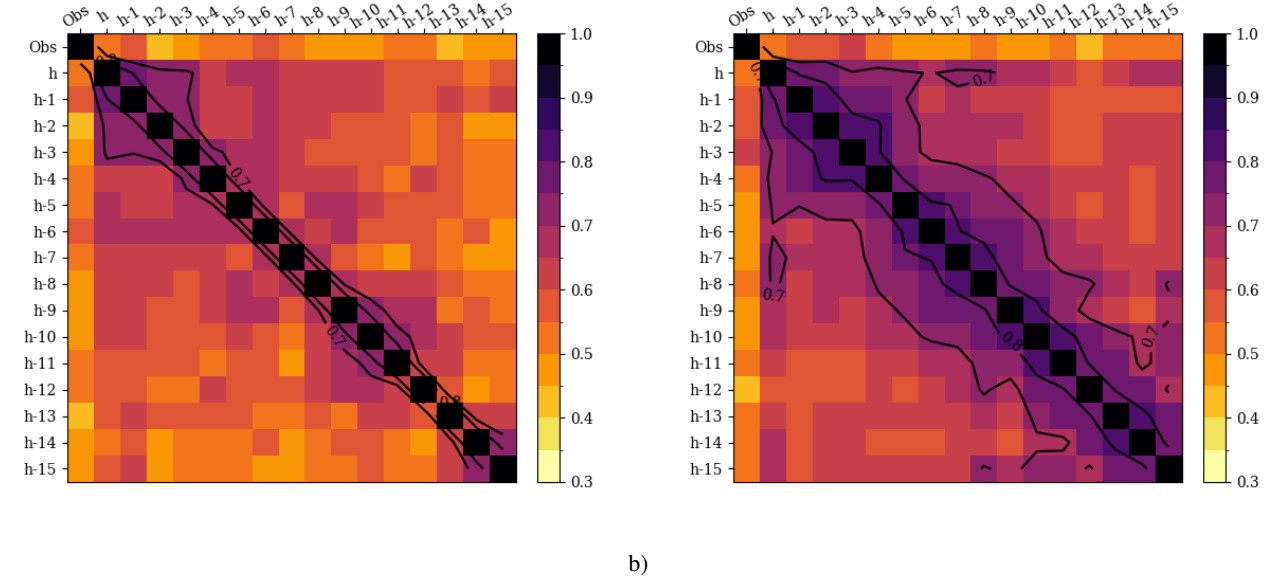

a)                                                                    b)

**Figure 13.** Correlations between accumulated precipitation field observed (first column : ANTILOPE blending product) or simulated by the different hourly initialized forecasts (h to h-15 columns) valid at the same time for 3dv (a) and 3devi (b) experiments averaged on the 8 HPE observed and simulated on different areas for the 19 Sep. 2020, the 03 Oct. 2020, the 09 Sep. 2021, the 14 Sep. 2021, the 25 Sep. 2021, the 3 Oct. 2021 and the 4 Oct. 2021.

increase when the forecast gets more and more recent (from h-15 to h) appears somewhat more clearly in 3devi than in 3dv. Moreover, correlations between the different forecast fields (columns h to h-15) appear to be larger in the 3devi experiment for all the analysis times considered, regardless of the delay between successive analysis times. For example, correlations between forecasts initialized by 3 hr lagged analyses (3rd off-diagonal lines) are always larger than 0.65 in 3devi, while the correlation

between forecasts initialized by 1 hr or 2-hr lagged analyses (1st and 2nd off-diagonal lines, respectively) rarely reach this value in 3dv.

## 6    Conclusion

After fifteen years of operational runs, a major update of the AROME-france DA scheme occurred in October 2024 : the historical 3D-Var scheme has been replaced by 3DEnVar, involving fully flow-dependent background error covariances, which

has been achieved after several years of thorough studies and experiments. While preliminary investigations have already been described in MO18 and MI21, the purpose of the present paper is to describe the set of studies which have been conducted in order to achieve this operational implementation.





The first part describes the complex current operational NWP system used at Météo-France in which the new scheme has been included, paying particular attention to its interactions with existing LAM EDA and EPS applications. Operational constraints require that the AROME-France 3DEnVar uses forecast perturbations from the AROME EDA in an asynchronous way : H-1, H and H+1 3DEnVar analysis times use 5, 6 and 7-hour range perturbed forecasts from the H-6 EDA analysis time.

The second part of the article is devoted to fine-tuning the different elements of the 3DEnVar scheme itself. Several sensitivity studies were performed and evaluated using diagnostics relevant for the DA cycle. Among these various elements, the horizontal localization length scale has been found to have the largest impact. Best performances were obtained with a regularly varying length scale along the vertical, from 25 km at the lowest model level, to 150 km at the highest vertical level. This optimal setting is different from those diagnosed in MO18 (170 km constant over the vertical) or resulting from sensitivity experiments in MI21 (40 km constant over the vertical) in different experimental frameworks. The vertical localization length scale setting is more consistent with these two previous studies, since the value of 0.3 ln(hPa) has also been retained here. Sensitivity experiments to the EDA spread were also performed, by modifying the multiplicative inflation factor. The purpose of such a modification would be to compensate possibly for the use of 5, 6 and 7-hour forecast ranges from the EDA to estimate the 1-hour background error covariances of the hourly DA cycle of the deterministic system ; in the end, the obtained optimal value of 1 confirms that the spread of the ensemble of perturbations used is quite correct. As in MO18, the use of a full ensemble-based B matrix (no hybridization) was also found to provide the best results. This is very important as it makes further developments easier, such as the extension of the control variable to new variables (hydrometeors, non hydrostatic variables) or to the temporal dimension (4DEnVar), since relevant flow-dependent background error covariances are directly available from the EDA. Finally, the use of IAU has been evaluated in order to reduce the spin-up, which is higher in 3DEnVar experiments than in the operational 3D-Var.

As a result of these investigations, a configuration, declined in two versions - without and with IAU -, has been retained and evaluated over a long period and for different meteorological events. Several averaged scores (against IFS analysis, radiosondes, surface stations) indicate that 3DEnVar outperforms the operational 3D-Var for all parameters in the troposhere at numerous forecast ranges. An outstanding example is the RMSE improvement of relative humidity at 850 hPa against IFS analysis (resp. radiosondes) reaching 8.7% (resp. 7.4%) at 12-hour forecast range, 4.1% (resp. 3.7%) at 24-hour forecast range and 2.5% (resp. 1.5%) at 36-hour forecast range. The simulation of different weather events is significantly improved with 3DEnVar. Results for rainfall and wind gusts are also better. The improvement of the official performance indicator of the AROME-France NWP system, mixing the two previous elements, reaches 1.1% during the winter period and 2.75% during the convective one.

In addition, the simulation of the strongest wind gusts during 6 winter storms is not affected by the higher spin-up in 3DEnVar. Finally, meteorological events sensitive to small scales, such as winter fog (16 IOPs in the SOFOG3D campaign) or Mediterranean HPEs (7 cases during the autumns of 2020 and 2021) are better simulated thanks to the 3DEnVar analyses. Regarding the simulation of these HPEs, the run-to-run consistency is also improved using 3DEnVar.

The initial conditions provided by this 3DEnVar scheme, perturbed thanks to the AROME-EDA information, also lead to significant improvement in the performance of the AROME-France EPS (PEARO) system. In addition, the improved performance



of the AROME-France short-range forecasts (2 to 7 hours) used as backgrounds in the nowcasting version of AROME-France (AROME-NWC) also has a positive impact on the performance of the latter (personal communications).

These results have allowed the 3DEnVar to become the new version of the DA scheme for the operational AROME-France NWP system. The retained configuration exhibits a good numerical stability after having run for more than a year. Yet the heat waves over France in the summer 2022 revealed some very rare weaknesses : four 1-hour forecasts of the DA cycle crashed as very strong temperature increments appeared in some points of an Alp Valley in the two lowest model levels, due to very narrow vertical background error correlations. For this reason, the IAU flavour of the defined configuration has been finally preferred, as it simulates this period without any problem, leading to its operational implementation on 15 October 2024.

This major change, combined with the use of OOPS software, offers very interesting prospects in terms of development. On one side, the scheme itself can benefit from the use of a Scale Dependent Localization (Caron et al., 2019) which is a way to introduce some flow dependency in the specification of the localization length scale : different localization length scales are allowed and are more or less active, according to the scales involved in the meteorological situation. Increasing the ensemble size can also be investigated using lagged- (Caron et al., 2019) or shifted-approaches in order to reduce the impact of sampling

noise. Adding hydrometeors in the control variable (Destouches et al., 2023) opens the way for investigations related to direct assimilation of radar reflectivity, in order to replace the current retrieval approach, as well as lighting observations (Combarnous et al., 2024). A major improvement is also expected from the shift towards a 4DEnVar system currently under development in order to assimilate radar, SEVIRI radiances, and surface station observations every 15 minutes in the 1-hour DA cycle. In this case, the temporal dimension along the assimilation window is managed by using a 4D state built with the 3D states from the

non linear-model at the different timeslots, and using background error temporal correlations between these different 3D states directly estimated from the EDA. First investigations demonstrate that the 4D increment involving the five 3D increments along the 1-hour assimilation windows every 15 minutes is temporally consistent even at convective scale. Finally, coupled DA can also be envisaged : preliminary works concerning the ability of an AROME EDA coupled with a 1D Ocean Mixing Layer to estimate background error cross-correlations between variables for the ocean and the atmosphere have been initiated, and first

strongly coupled ocean-atmosphere analysis increments have been obtained.

*Code availability.* The source code of AROME-France, included in the operational ARPEGE/IFS code, cannot be obtained.

*Data availability.* The analyses and the forecast fields of the operational AROME-France model are daily available in the Météo-France open data database https://donneespubliques.meteofrance.fr/

*Author contributions.* Valérie Vogt and Pierre Brousseau prepared and carried out all the numerical experiments used in this study. They
investigated the results and wrote the paper with the help of all the coauthors. Etienne Arbogast provided help in the technical implemen-



tation of AROME-3DEnVar in OOPS, in connection with OOPS-ARPEGE developments. Maud Martet and Guillaume Thomas helped to investigate the results by performing diagnostics and verification computations. Loïk Berre contributed more specifically to some parts of paper writing and proofreading

*Competing interests.* The authors declare that they have no conflict of interest.

*Acknowledgements.* The authors would like to thank Laure Raynaud and Gregory Roux (respectively Nicolas Merlet and Thibaut Montmerle) for their evaluation of the impact of the AROME-France 3DEnVar analyses, as initial conditions (respectively subsequent forecasts as backgrounds) in the AROME-France EPS (PEARO) (respectively AROME-France nowcasting) system.



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
