# Peer review of "The operational 3DEnVar data assimilation scheme for the Météo-France convective scale model AROME-France"

_EGUsphere, 2025_

## Referee Comment (RC1)

Review for:   The operational 3DEnVar data assimilation scheme for the Meteo-France convective scale AROME-France,

by Pierre Brousseau, Valerie Vogt, Etienne Arbogast, Maud Martet, Guillaume Thomas, Loik Berre

General vote: I recommend accepting the manuscript after minor revisions.

General comments:

The Manuscript presents the work carried out to define the operational configuration of a new data assimilation scheme for the convective-scale model AROME at Meteo-France. This includes a number of sensitivity tests as well as the description of the operational configuration. It is shown that the new scheme based on 3DEnVar, using flow-dependent ensemble-based instead of climatological background error covariances, outperforms the old operational 3DVar scheme both in terms of statistical scores and for different kinds of individual severe weather events.

While the manuscript does not directly address a better understanding of natural hazards, it does provide a significant contribution for an improved operational prediction of such hazards. 3DEnVar is a state-of-the-art method also used at other centres for both global and regional data assimilation, and the sensitivity experiments provide relevant information and valuable insight in its application at the convective scale.

Generally, the work has good scientific originality, high relevance and quality, and the manuscript is well-written, well-referenced, and well-structured.

Minor specific comments:

Line 203: "… all sensitivity tests will be compared using the diagnostics that have been found to be representative of the obtained results." These diagnostics only include results from the 1-hourly DA cycle. In the end however, the purpose of the DA in the context of operational NWP is to provide initial conditions which result in the best forecast quality (in the case of AROME-France with lead times > 3 h since shorter lead times are available in real time only after their validity time). I think, the decision to use only diagnostics from the DA cycle for all the sensitivity experiments (except for lines 297 – 298) deserves more explanation here.

Lines 211 – 213: I agree, simulated precipitation, being the result of many processes, can be a good indicator of the general behaviour of the system. However, (sums of) 1-h precipitation accumulation in the 1-h DA cycle may be strongly affected by (short-lived) spin-up effects. These may or may not have a significant impact on the forecast quality at later lead times. (In Section 3.3, e.g., the spin-up seen in surface pressure tendencies appears to have only a limited impact on the forecast quality, at least in the sense that experiments 3dev and 3devi perform fairly similar.) From that point of view, I would think that these 24-h precipitation sums of the DA cycle can be a valuable, but possibly quite incomplete indicator of the system behaviour with respect to precipitation. This should also be addressed (perhaps simply by further explanations, e.g. that the authors also looked at precipitation at longer lead times, but found similar results in the sensitivity tests (i.e. when comparing experiments) as in the diagnostic shown).

Lines 200 – 201: "… the localization length scale appears to be the most important parameter to investigate first." In my view, it need not be the most important parameter to be investigated and tuned first. It could also be the parameter for which the decision is most clear. I am thinking here of the application of IAU which is shown to greatly reduce

imbalances leading to spin-down of surface pressure tendency. Even though the impact of this on the forecast scores is shown to be quite limited (mostly slightly positive, but mostly not statistically significant), I would see this reduction of imbalances as a benefit on its own, which in the end proved to avoid the rare crashes of model simulations occurring without IAU. Hence, in the end, the chosen operational configuration includes IAU. This point may be relevant in the context of lines 268 – 273, e.g.: "The better results obtained in this study with shorter localization length scales than the optimal ones diagnosed with Ménétrier et al. (2015) could be explained by the fact that, while filtering sampled covariances, the localization helps to filter the numerical noise due to imbalances in the analysis and thus improves performances of the hourly DA cycle used. …". These mentioned results in Section 3 were obtained without IAU, and my question is, whether this is also true to the same extent with IAU? In other words, would the optimal length scales be the same or larger when IAU is applied? Did the authors perform also sensitivity tests on the optimal localisation length scales with IAU?

Lines 260 – 262: "Spin-up can also be referred as overestimated precipitation in the first few hours …. This aspect has already been investigated in section 3.2 and illustrated in figure 3…" This is not strictly true, because e.g. figure 3 shows HSS which is a skill score and not a measure of the bias which would be needed as an indicator of spin-up (if it can be referred as overestimated precipitation). Furthermore, it could be clarified that (a) 'overestimated' should be understood here in comparison with longer lead times, not in comparison to observations, and (b) underestimation (in this sense) could also be a result of spin-up effects.

Section 5.3.3, figure 13: Very interesting!

Technical comments:

Line 24: "using the Berre (2000) formulation"

Lines 27 – 28: FGAT: did this imply that observations were used at higher resolution. Any references?

Line 38: "allow the numerical cost"

Line 55: replace "counterpart" by e.g. "downside" or "drawback"

Line 73: "A hybrid"

Line 93: possibly better to replace "general scores" by "statistical scores"

Line 94: I do not understand what is meant by "original" approach

Line 97: "numerical weather prediction system"

Line 112: This implies that PEARO runs at 1.3 km, while the EDA is at 3.2 km. It is worth mentioning how exactly the initial conditions of PEARO are obtained. Important to know also for lines 555 – 556.

Line 127 – 128: for a better understanding: "These long forecasts (denoted 'production' in figure 1) have …"

Line 131: I would think: "the IAU is not only used for its filtering purposes, but it is also employed to update the forecast with more recent observed information." More importantly, it should be clarified in Section 3 and 4 that H+1 observations and H+1 analysis increments (line 129) are not used in the sensitivity experiments with IAU (experiments "IAU" respc. "3devi") in order to allow for a fair comparison with the other experiments (or are they used?). I.e. that the same set of observations are input to all the sensitivity experiments.

Line 141: replace "H+3" by "H-3" (?). H+3 would imply that the 9 UTC AROME-EDA uses 0 UTC AEARP lateral BC which would contradict figure 1.

Line 153: "… and while different uncompleted …"

Figure 1: Elsewhere, the AROME EDA is named PEARO instead of AEARO.

Figure 1: For clarity, please explain "guess" somewhere.

Line 155: " … ARPEGE DA system based on the OOPS code with all the ingredients …" (?)

Line 185: "control variable" (or rather: control vector?): should be explained what it is.

Line 185: "..- two horizontal wind components…"

Figure 2 caption: "… indicate larger (resp. smaller) RMS values …"

Lin 235: "… show very similar performances …"

Line 281: T, q, Ps have not yet been defined. Could be defined e.g. on line 186.

Figures 5 caption: remove "specific or"; in Figures 5 and 7, I would replace "Bold numbers" simply by "Numbers", because in Figures 2, 3 one has to distinguish between normal (not bold) numbers and bold numbers, while in Figures 5, 6, 7 when it says "bold numbers" I am trying to distinguish between bold and normal numbers but I cannot find any normal numbers.

Figure 5 and 6: I do not understand how a difference of 0.0 can be a statistically significant difference (by bootstrap). Please explain how this can happen, or what it means.

Line 353 / Figure 6: "…IAU … improves the analysis fit to observations." Do you have any idea why this is? Does it have to do with spin-up effects? What about the RMS Diff to the first guess (1-h forecast) which we cannot see in figure 6?

Figure 6, figure 2: The application of IAU is shown to lead to clearly smaller O-A against radiosondes in Figure 6, but hardly smaller O-A against aircraft data in Figure 2. What is the reason for this difference? Is it the different verifying observations? Or the different period? Or are there relevant discrepancies between (a) the differences between experiments 3devi and 3dev (in Figure 6) on the one hand and (b) the differences between experiments IAU and Loc h25-150 on the other hand? Please address this also in the manuscript.

Line 358: Maybe it is worth to briefly explain "double penalty". Something like: " … which affects classical local scores if e.g. small-scale phenomena like convective cells are predicted well but with some dislocation." (Or provide a reference.)

Line 359: replace "excedence" by "exceedance"

Line 363: For clarity, I would write: "This neighbourhood is used to compute the AROME-France Performance Index employed to measure the evolution of the quality of the AROME-France forecasts. To this end, the index averages BSS values of 0.5, 2.0, 5.0 mm/6hr precipitation and 40 km/h wind gust for the 6, 12, …" (Otherwise, one has to re-read the whole paragraph to understand what could be meant by "AROME France Performance Index" in lines 370 – 371.

Line 408: please explain why no forecasted starting during the daytime are evaluated. Have there been no differences between the 3 experiments for these forecasts?

Line 410: better to replace "…(IOPs) were planned … " by " …(IOPs) were carried out …" (?)

Lines 415 – 416: replace "… closer / higher … better …" by " … the closer / the higher … the better …"

Line 456: "… the general better accuracy of the location of the event in the 3devi experiment."

Line 490: "location error" instead of "error location"

Lines 507 – 508: I would write: " … forecasts (from H, the most recent, to H-15, the oldest), all valid at the same time, …"

Line 550: "mixing the two previous elements". Maybe easier to digest: "which accounts for the forecast quality of precipitation and wind gusts"

Line 555 – 556: Why does it lead to an improvement of PEARO, given that the EDA still relies on 3DVar and hasn't changed? See comment for line 112.

Line 552: "Alpine Valley"

Line 570: "lightning"

Lines 576 – 577: Wouldn't using 5 time slots every 15 min in a 1-hour DA cycle imply that the observations in the first resp. last time slot are used twice in the DA cycle? Please comment.